# Comparative Modelling of Organic Anion Transporting Polypeptides: Structural Insights and Comparison of Binding Modes

**DOI:** 10.3390/molecules27238531

**Published:** 2022-12-03

**Authors:** Arun Kumar Tonduru, Santosh Kumar Adla, Kristiina M. Huttunen, Thales Kronenberger, Antti Poso

**Affiliations:** 1School of Pharmacy, Faculty of Health Sciences, University of Eastern Finland, P.O. Box 1627, 70211 Kuopio, Finland; 2Institute of Organic Chemistry and Biochemistry (IOCB), Czech Academy of Sciences, Flemingovo Namesti 542/2, 160 00 Prague, Czech Republic; 3Department of Internal Medicine VIII, University Hospital Tuebingen, Otfried-Müller-Strasse 14, 72076 Tuebingen, Germany; 4Department of Pharmaceutical and Medicinal Chemistry, Institute of Pharmaceutical Sciences, Eberhard-Karls-Universität, Tuebingen, Auf der Morgenstelle 8, 72076 Tuebingen, Germany; 5Cluster of Excellence iFIT (EXC 2180) “Image-Guided and Functionally Instructed Tumor Therapies”, University of Tübingen, 72076 Tuebingen, Germany; 6Tuebingen Center for Academic Drug Discovery & Development (TüCAD2), 72076 Tuebingen, Germany

**Keywords:** organic anion transporting polypeptides (OATPs), molecular dynamics simulations, homology models, xenobiotic transporters, Solute-Carrier O (SLCO)

## Abstract

To better understand the functionality of organic anion transporting polypeptides (OATPs) and to design new ligands, reliable structural data of each OATP is needed. In this work, we used a combination of homology model with molecular dynamics simulations to generate a comprehensive structural dataset, that encompasses a diverse set of OATPs but also their relevant conformations. Our OATP models share a conserved transmembrane helix folding harbouring a druggable binding pocket in the shape of an inner pore. Our simulations suggest that the conserved salt bridges at the extracellular region between residues on TM1 and TM7 might influence the entrance of substrates. Interactions between residues on TM1 and TM4 within OATP1 family shown their importance in transport of substrates. Additionally, in transmembrane (TM) 1/2, a known conserved element, interact with two identified motifs in the TM7 and TM11. Our simulations suggest that TM1/2-TM7 interaction influence the inner pocket accessibility, while TM1/2-TM11 salt bridges control the substrate binding stability.

## 1. Introduction

Organic Anion Transporting Polypeptides (OATPs) are membrane transporters that play an essential role in the uptake of several endogenous substances, such as hormones, bilirubin and bile salts, and exogenous compounds, such as xenobiotics, drugs and toxins [1]. These OATPs are implicated in different cancers, such as colon, pancreatic, gastric, lung, breast [2] and prostate cancers, and dysregulated in Chron’s disease, primary sclerosing cholangitis, and kidney and liver diseases [3]. Moreover, several studies highlighted the role of OATPs in the pharmacokinetics of a wide range of drug substances, such as statins, anti-hypertensives, antifungals, antivirals, anticancer agents and cholesterol-lowering drugs [3].

OATPs belong to the Solute-Carrier O (SLCO) protein family expressed in various tissues including liver, intestine, kidneys, testis, eye, placenta, lungs, brain (blood–brain barrier, endothelial cells and astrocytes), and some OATPs (such as OATP2A1 and OATP4A1) are ubiquitous [4]. OATPs such as 1B1 and 1B3 are highly expressed in the liver and influence the uptake of several drugs into the blood and their involvement in the drug–drug interactions [5]. Moreover, the United States Food and Drug Administration (US FDA) and the European Medical Agency (EMA) recommend the assessment of the new investigational drugs against OATP1B substrate/inhibitor using in vitro assays [6,7].

In humans, eleven types of OATPs have been found that include OATP1A2, 1B1, 1B3, 1C1, 2A1, 2B1, 3A1, 4A1, 4C1, 5A1 and 6A1 [8]. Classification of OATPs is based on the percentage identity of amino acid sequence within the superfamily (OATP/*SLCO*), and they are subdivided into families (>40%, e.g., OATP1 and OATP2), subfamilies (>60% OATP1A and OATP1B) and individual genes (e.g., OATP1B1 and OATP1B3) [9]. OATP1A2 (SLCO1A2) was the first characterized OATP transporter in humans distributed widely in the human body, mainly in liver cholangiocytes, retinal tubules, salivary glands, testes and most importantly endothelial cells of the brain, which compose the blood–brain barrier (BBB) [10]. Hence, to improve the drug absorption and availability to the targeted tissue majorly BBB, OATP1A2 mediated drug delivery has been studied as an approach for delivering drugs to the central nervous system [11,12,13]. However, a complication of this method is that OATP1A2 has a wide range of natural substrates including bile salts, bile acids, thyroid hormones, steroid conjugates and unnatural substrates including drugs, such as methotrexate, fexofenadine, erythromycin, imatinib, lopinavir and some β-blockers [13]. Interestingly, the δ-opioid receptor agonists Deltorphin II, D-penicillamine, enkephalin and triptans are the only known OATP1A2 substrates that can cross the BBB (Figure 1). There is little known about the binding mechanism of these diverse molecules to OATP1A2 [14]. The OATP1A2 transport mechanism is sodium independent, but the driving force is not known. Thus, it is essential to understand how the OATP1A2 is binding to diverse substrates.

While OATP1A2 is the most studied of all OATPs, there is also some sparse data concerning other OATPs’ substrates/inhibitors (Figure 1). HMG-CoA reductase inhibitors, ACE inhibitors and angiotensin II receptor antagonists are substrates of OATP1B1. OATP1B1 is expressed in hepatocytes and small intestine enterocytes, where it plays a key role in drug–drug interactions. In vivo studies have proved that drugs, such as cyclosporine and gemfibrozil, affect the bioavailability of other drug substrates of OATP1B1 (namely, statins, repaglinide and bosentan) through OATP1B1 inhibition [15]. Cyclosporine is also reported to inhibit OATP1B3, which is majorly expressed in the liver. However, OATP1B3 transports digoxin and paclitaxel and some other drugs [16,17]. OATP2B1, which is expressed in the liver (sinusoidal membrane of hepatocytes) and heart, was reported to be inhibited by compounds cyclosporine and gemfibrozil [17]. Other transporters, OATP1C1, OATP2A1, OATP3A1 and OATP4A1, may also have the potential to be involved in the drug disposition [15]. However, focused studies are not available concerning their mechanisms related to drug transport or kinetics. This knowledge would be needed to design specific substrates or inhibitors targeting these transporters that will facilitate new opportunities for drug design and interpretation of already known drug–drug interactions.

Currently, there are no OATPs crystal structures available, which would allow inspecting the binding of these molecules. Computational approaches (such as hydropathy analysis) have revealed that all members of OATPs have 12 transmembrane helices (Figure 2) with N- and C-terminals facing the cytoplasm. Structures include a large conserved extracellular loop, between 9th and 10th transmembrane helices, with ten conserved cysteines [18,19]. These cysteines are needed for surface expression and might form disulphide linkages [20]. All OATPs have a signature amino acid sequence (D-X-RW- (I, V)-GAWW-X-G- (F, L)-L) (where X can be any amino acid) between extracellular loop 3 and the transmembrane domain 6 (Figure 2A). Site-directed mutagenesis experiments revealed the importance of signature sequence in surface expression of the protein in OATP1B1 [21]. OATPs are found in both non-glycosylated and glycosylated forms with molecular weights ranging from 60 to 150 kDa [22]. Glycosylation occurs on the 2nd and 5th extracellular loops through the asparagine residues (for example Asn134, Asn503 and Asn516 in OATP1B1) affecting the stability and function of the transporters [15,22].

Besides this data, several studies have reported comparative models for OATPs, whose quality and information level greatly varies with the selected template [23,24,25,26,27,28,29,30,31,32]. For instance, Nan li et al. generated an OATP1B1 homology model based on glycerol-3-phosphate transporter as template (PDB ID: 1PW4, from *E. coli*) to interpret mutagenesis data on transmembrane helix 2 [25]. Similarly, Hong et al. generated an OATP1B1 homology model to study the mutagenesis effects on TM11 [26]. Mendery et al. compared the conserved positively charged residues in the entrance of the pore among OATP1A2, 1B3 and 2B1 [23]. Glaeser et al. constructed a homology model of OATP1B3 to study the conserved amino acids Lys41 and Arg580 [27]. Several other studies have generated homology models with most of them having only transmembrane regions. Most of these models lack some extracellular loops, due to the low similarity with templates. These studies were mainly focused on using models to interpret the single point mutation data for a small set of transporters. Nevertheless, a more comprehensive comparative study discussing the substrate-binding mechanisms remains is needed.

In this study, we reported the comparative models of OATPs and explained the binding modes of substrates using docking and molecular dynamics (MD) simulations. To construct OATP models, we have used the multi-template homology modelling technique due to the low sequence identity in the loop regions. In addition, we compared the modelled OATP structures with one another to theoretically address their structural diversity. Our models are freely available (https://zenodo.org/record/7071180) (generated on 12 September 2022) and can be used in future OATP research to design new ligands and to help in interpreting the chemical biology studies.

## 2. Results and Discussion

### 2.1. OATPs Share a Conserved Transmembrane Helix Fold Harbouring a Druggable Binding Pocket

Due to the lack of three-dimensional structures of OATP’s, theoretical structure prediction methods became the main source of information for the structure-based drug design and interpretation of mutational data of OATPs. The critical challenge in the modelling of OATPs is the low sequence similarity against the available templates. Here, homology models of all 11 OATP subtypes (1A2, 1B1, 1B3, 1C1, 2A1, 2B1, 3A1, 4A1, 4C1, 5A1 and 6A1) were generated using multi-template modelling technique. The homologue search resulted in the identification of bacterial D-galactonate proton symporter (PDB ID 6E9N, chain A) as the top hit with 9–12% global sequence identity (accessed on 2 March 2020). This symporter belongs to the Solute carrier (SLC) transporter superfamily and was selected as the main template to build the OATP models. Due to the importance of the 5th extracellular loop in glycosylation and the function of substrate transport and surface expression, this loop region was modelled using another template homologue: the pancreatic secretory trypsin inhibitor (Kazal type, PDB ID:1TGS, Chain I). Sequence identities and homology of each OATP to the templates are presented in the Appendix A: Table 1. A recent search in BLAST gives a better template Kazal-like domain containing protein Slco6c1 (7EEB Chain L) of mouse with 21% identity and query coverage of 94% [28].

All the 11 OATPs are predicted to have 12 transmembrane helices (TM, Figure 2A), such as the template, with a pore open towards the cytoplasm with six extracellular loops (ECL) and five intracellular loops. All transmembrane helices, with exception of TM3, TM6, TM9 and TM12, face the central axis and to some extent compose a pore (Figure 2B). The helices TM3, TM6, TM9 and TM12 are far from this central pore region, directly interacting with the membrane. The extracellular loop 5 (ECL5, Figure 2A,B) present between the TM9 and TM10 had its 3 disulphide bridges preserved to keep the same fold as that of the template. N and C-terminal ends are positioned in the cytoplasmic region. Finally, the N-glycosylation sites were found in the second and fifth extracellular loops for some OATPs, based on the multiple sequence alignment and literature [22,28,29] also highlighted in Appendix A. Our homology models were compared with the Alphafold2 models and the RMSD along the transmembrane helices ranges from 1.97 to 3.19 Å, and the results were reported in the Appendix A.

### 2.2. Binding Site Analysis and Classification 

OATPs’ potential binding sites were predicted using SiteMap and ranked according to their SiteScore values, where SiteScore values higher than 0.8 stand for an acceptable druggability score [30]. For all the models, this value ranged from 1.0 and 1.23 (Appendix A), and the highest scored pockets were identified in the central pore region, encompassing the inner interfaces of transmembrane helices (Figure 3A). Visual examination of the druggable sites of all OATPs revealed high sequence similarity (ranging 30–80%) among them. However, more calculations of identity and similarity matrices (Appendix A) using the alignment of the binding sites (Figure 3B) supplied details of specific substitutions among the OATPs, resulting in three clusters (Figure 3C). From the matrices and clustering, it was revealed that OATP1A2 is 55% identical or 69% similar to OATP1C1 in the pore region and a lesser extent to the same sub-family members (OATP1) 1B1 (47%) and 1B3 (33%) and falls in the same sub-cluster. The pore region of OATP1B1 and 1B3 showed 69% identity, 83% similarity and transport common substrates except for cholecystokinin octapeptide -8 (CCK8) and telmisartan, which are selective to OATP1B3. From class 2, it was revealed that OATP3A1 showed 33% and 38% identity with 2A1 and 2B1, respectively, and falls into the same cluster. OATP2A1 and 2B1 show 61% identity or 83% similarity along the pore. 

Moreover, these three transporters in class 2 transport common substrates such as alprostadil (PGE1) and dinoprostone (PGE2). In class 3 transporters OATP4A1 and 4C1 show 47% identity and are clustered in the same class along with 5A1 and 6A1. OATP4C1 and OATP6A1 show the lowest identities with OATP1 and OATP2 families and the least with OATP1A2 (8.3% with 4C1) and 5.5% with 6A1. Interestingly, the OATP6A1 binding site shares the highest identity (38.9%) only with OATP4C1; however, little is known about the 6A1 transporter.

### 2.3. Electrostatic Potential and Hydrophobic Surface of Pore

The pore diameters of all the OATPs (Appendix A) were calculated and show a similar trend where the diameter is wide towards the intracellular region and narrow towards the extracellular region reflecting the inward open models for all the OATPs. The electrostatic potentials provide information on the nature of the substrate’s recognition. Here, electrostatic complementarity maps were adjusted to view only the surface of the putative pore as seen in Figure 4. OATP1 family or class 1 (Figure 3) shows both positive and negatives potentials except for 1B3, which shows more positive electrostatic potential than the other three transporters (1A2, 1B1 and 1C1) Appendix A. In the OATP2s (2A1 and 2B1) family, the pore region displays more positively charged surfaces than the rest of the transporters, and just a small patch of negative potential was observed towards the intracellular region in 2A1. The pore region of both OATP1B3 and 2B1 has shown positive electrostatic potential similar to previous studies [31]. Inversely, OATP3A1 shows negatively charged potential in the middle of the pore and positive potential towards intracellular region. In class 3, all four transporters show negative potential surfaces throughout the pore, except 5A1, which shows positive potential towards the intracellular region. Overall, the electrostatic potentials of the pore region indicate the amphipathic nature of the pore positive/negative in class 1 that allows the transport of both negatively and positively charged substrates. Positive potentials in class 2 and negative potentials in class 3 OATP transporters point out hydrophobic and polar nature of the pores, respectively. Additionally, the hydrophobic volumes, calculated using the pore lining amino acids, agreed with the above predictions. The OATP1 family has low hydrophobic volumes (Appendix A) compared to the other families of OATP. OATP1B3 shows higher hydrophobic sites than the other three transporters in the same family (OATP1) by accommodating the residues. Overall OATP3A1 shows the highest hydrophobic volume among the transporters. Class 2 and class 3 transporters show high hydrophobic volume compared to class 1. These results correlate with the MD simulation derived interaction profiles in which class 2 shows high hydrophobic interactions, and class 1 shows high polar interactions. From the electrostatic potentials and hydrophobic volumes, OATP1B3 shows more positive electrostatic fields and hydrophobic volume than the others in the same family OATP1, while 1B1 shows the lowest hydrophobic volume among all OATP transporters.

### 2.4. OATP Substrates Display Multiple Binding Modes with a Dynamic Interaction Profile

The highest scored druggable site was found to be in the central pore region, encompassing the inner interfaces of transmembrane helices. This site of OATPs was used for docking of all the studied compounds (Figure 1). Visual inspection of docking results showed a separation of ligand conformations into two distinct poses (from here on referred to as Pose1 and Pose2). 

As an illustrative example, the substrate estrone-3-sulfate (E3S) potential binding mode in the different OATP models and their interactions are detailed below. E3S-Pose1 (Figure 5A) represents the conformation where the E3S steroid ring is vertically bound along the pore, with the sulphate group at C-17 facing the extracellular region, and the carbonyl group at C-3 position facing the intracellular region (Figure 3A). This pose was ranked as the topmost pose in OATP1A2, OATP1C1, OATP2A1, OATP2B1 and OATP4A1. The most common interactions in pose1 are between sulphate oxygens of E3S and the conserved arginine in TM11 and lysine from the TM1 (Lys33 in OATP1A2, Lys41 in OATP1B3, Lys56 in OATP1C1, Lys53 in OATP2A1 and Lys70 in OATP2B1). Pose2 (Figure 5A) is observed in OATP1B1, OATP3A1 and OATP4C1, and it is a vertical flip of the Pose1, with the sulphate group facing the intracellular exit of the transporter. The sulphate group in pose2 is placed similarly to the pose1 which is found near the cluster of positively charged residues by keeping stable hydrogen and ionic interactions with conserved arginine in TM11. 

The two potential binding modes for the respective OATPs were subjected to at least 600 ns (3 × 200 ns) MD simulations. Both docking poses demonstrated the interconvertibility of the docking poses (Pose 1 and Pose 2) during the simulations, where Pose 1 was observed to be flipping to form Pose 2 (Figure 5B). However, the interactions were consistent with the conserved amino acids. During the trajectory analysis, we also noticed that the substrate (E3S) was moving from the extracellular region towards the intracellular region resembling the substrate transport. In OATP1B1, the pose is vertically positioned downwards to the intracellular region. There is slight variation in the pose of E3S in OATP1B3 as it is slightly horizontal in position, and it variates between horizontal and vertical poses during the simulations. The poses of OATP2A1 and OATP2B1 are slightly further up towards the extracellular region than the other poses. The flipping of the pose, inter-conversion from horizontal to vertical pose and movement of the substrate towards the intracellular region could supplement the knowledge of two binding sites and transport mechanisms in most of the OATPs. 

Earlier reports on docking of steroids showed the orientational versatility within the binding sites [32] since the 3- and 17- carbon substituents form most of the key interactions for stabilization. Previously reported docking poses of strong OATP1B1 inhibitors (E3S, lithocholate and cholic acid methyl ester) and OATP1B3 binders (digoxin, lithocholate and cholic acid methyl ester) showed common orientations in the central binding cavity [33]. OATP1B1 ligands appear to be vertically oriented, while OATP1B3 featured a horizontal orientation but with a less consistent binding pattern. Interestingly, their work also suggests that the salt bridge Lys49-Glu74 (TM1-TM2, respectively, in OATP1B3) restricts the N-terminal pocket volume favouring the horizontal orientation of the ligands. The interaction of residues with E3S along the pore observed in MD simulations can be seen in the bubble graph (Figure 5C). As seen in the bubble figure, the interactions among the OATPs in one cluster are similar; for example, hydrogen bonding interactions are observed more in class 1 than the others; hydrophobic interactions are predominant in class 2 (Appendix A). In class 1 (at least 3 OATPs) Lys33, Arg556, Glu66, Leu176, Glu200, Ala203, Gln328, Phe332, Ile529, Thr552 and the corresponding residues interact with E3S. Leu176, Glu200, Ala203, Phe332, Arg556 and the corresponding residues in other OATPs have shown interactions along the simulation with E3S in all OATPs, with some exceptions. 

Previous studies from Li et al. (2012) demonstrated a combination of modelling and mutagenesis experiments that several TM2 residues of OATP1B1 (namely, Asp70, Phe73, Glu74 and Gly76) are implicated in transport function [25]. Moreover, using kinetics experiments, two distinct binding sites (low- and high-affinity binding sites) for E3S were proposed, which agrees with our current model simulations proposing multiple sub-pockets within the OATP family.

### 2.5. Conserved Salt–Bridge Interactions’ Potential Role in Controlling OATPs Entrance

Multiple sequence alignment revealed the conserved amino acid residues among the OATPs that play a vital role in the binding or transport of substrates through the pore region (Appendix A). The relevant conformation and interactions of these residues were further investigated during our simulations.

There is a conserved sequence (ERR/ERQ/EKR) on TM1 extending partly to the extracellular loop (ECL1), which displays one negatively charged and two positively charged residues among almost all the OATPs (Figure 6A,B). This motif interacts with another conserved motif (KXXEQQ/KXXEXQ, where X can be any amino acid) on TM7 and conserved Ser/Thr on TM2 altogether forming the pore entrance (Figure 6A,B). We investigated their interactions and movement, due to their potential role in controlling the extracellular entrance. The glutamate on TM1 forms a hydrogen bonding network with Lysine on TM7 and Ser/Thr on TM2, which combined forms the pore entrance and the network. To explore the interaction stability across the simulations, we have measured the distances between the three residues and plotted them with time (Appendix A). The distance between Glutamate (TM1) and Lysine (TM7) was stable along the simulation with a median ranging from 1.6–5.8 Å. Most of the OATPs have shown the median at 1.7 Å except for OATP1C1 which has shown 5.8 Å (with E3S) and 3.5 Å (apo). Similarly, the distances between Ser/Thr on TM2 and Lysine on TM7 revealed that the median of the distances was high (above 5 Å) for some transporters (1B3, 1C1 and 3A1), and for others, it was below 5 Å, distances for the Ser/Thr on TM2 and Glutamate (TM1) (Appendix A). In general terms from the plots, we can see that the median distance in the OATP apo structure is lower than the median in E3S bound structures. We have seen that the network could be disrupted (Figure 6C), and the pore could open based on these interactions, which could allow the transport of the substrates. The interaction network is shown in Figure 6C with OATP1A2 displaying the network of interactions between Glu48 (TM1), Thr55 (TM2) and Lys341 (TM7) where the entrance of the pore looks closed. In Figure 6C, we can see the disruption of the network between these amino acid residues and the pore looks open. There is one more interaction between the conserved positive residue on TM1 (Figure 6C,D, Arg49) and conserved Glutamate on TM7 (Figure 6C,D, Glu344) which could also help in the entrance of substrates. The sequence motifs and interaction network are seen in all the studied OATPs, except OATP6A1 where the relevant lysine (TM7) (replaced by Isoleucine) and glutamate (TM1) (replaced by Glutamine) residues of the motif are not conserved.

It has been reported that the conserved Lys361 on TM7 of OATP1B3 is involved in the transport of substrates such as BSP and taurocholate [23] and participates in the recognition of substrates. In the same study, it was shown that the Lys361 is also crucial in substrate recognition of OATP1B1. Our analysis supports the above reports and reveals the importance of residues on TM1, TM2, and TM7 for the entrance of substrates and some recognition mechanisms. We propose that this interaction network can be relevant for regulating the extracellular entrance of ligands. Moreover, since the glutamate residues (on TM1 and TM7) are negatively charged at neutral pH, low pH’s might disrupt the interaction networks and open the channel. 

### 2.6. Conserved Lysine on TM1 and ‘RGI/MGE’ Motif on TM4 among OATP1 Family and Importance in the Ligand Binding

The TM1 conserved lysine present in all Class 1 (OATP1) family transporters (Lys33, 1A2; Lys41, 1B1; Lys41, 1B3; and Lys56, 1C1; Figure 7A,B) and their role in the transport of substrates have been established in previous studies [26,33,34]. This lysine shows ionic, hydrogen bond and water-mediated interactions with the ligands in all OATP1 family members during all analysed trajectory times. TM1′s lysine residue forms salt bridges with the adjacent glutamate from TM4 (Glu172, 1A2; Glu185, 1B1 and 1B3; and Glu201, 1C1), and this glutamate forms hydrogen bonds with arginine on the same helix (Arg168, 1A2; Arg181, 1B1 and 1B3; Arg197, 1C1). Complementarily, these TM4 residues compose a conserved motif (RGI/MGE) in the OATP1 family, where their side chains face towards the pore region and form stable polar interactions among themselves and/or with E3S, besides the TM1′s lysine. From the simulations, it is to be noted that the lysine shows interactions with E3S in all class1 OATPs more than 80% of the time except 1B1 and 1C1 (Figure 7C). Glutamate on TM4 forms water interactions with the ligands in class1 OATPs with 1B3 showing interactions for more than 70% of the time (Figure 7D). For instance, 17BES pose in OATP1B1 shows interactions with Glu 185 and Lys41, and E3S in OATP1C1 shows interactions with Lys56 and Glu201, along the simulations (Figure 7E). Curiously, in OATP1A2 mutation studies, it has been shown that interfering with Arg168 and Glu172 decreases methotrexate uptake, emphasizing their relevance to substrate transport [35]. Replacing Arg181 in OATP1B1 with histidine affected the affinity of estradiol-17β-glucuronide [33]. We propose that this interaction network would play a role in substrate binding and transport. 

### 2.7. OATP1′s TM2 Conserved Motif ‘GSFEIGNL’ Displays Stable Interactions with Conserved Arginine on TM11

Besides the conserved TM1/2-TM7 interaction, the TM2-TM11 interactions also seem to be relevant. The motif ‘GSFEIGNL’ is conserved in the OATP1 family and some residues such as glutamate/aspartate accompanied by an asparagine from this motif extend the conservation to other OATPs. For example, in OATP1A2 (Glu66, Asn69), OATP1B1 (Glu74, Asn77), OATP1B3 (Glu74, Asn77), OATP1C1 (Glu89, Asn92), OATP2A1 (Glu78, Asn81), OATP2B1 (Glu95, Asn98) and OATP3A1 (Glu86, Asn89). The glutamate residue is replaced with aspartate in 4A1, 4C1, 5A1 and 6A1, asparagine is conserved in 5A1 but not in OATP4 family.

The Glu/Asp on TM2 forms salt bridges and polar contacts with the conserved arginine on TM11 (Figure 8A). The interaction between these two residues is stable across the simulations in most of the OATPs investigated. Whenever present in the initial model, these interactions are then maintained during the entire simulation time (Figure 8A), and it seems to be ligand-independent, as both apo and E3S-bound simulations display similar frequencies for each model. As an illustrative example, we can see in OATP6A1 the interaction between Asp152 and Arg624 (in light pink, Figure 8C) also Glu66 and Arg556 in OATP1A2 (displayed in green). 

Our simulation data also show the polar contacts between the ligand and the Glu/Asp on TM2 (Figure 8B). Estradiol-17β-glucuronide shows hydrogen bond interactions with Glu74 for 30% of simulation time and water-mediated interactions for 60% of the simulation time in OATP1B1 (Figure 8B).

As mentioned, TM11′s arginine is conserved in all the OATPs, and its side chain faces the inner pore in all our models. This arginine displays stable polar contacts in either the ligand’s carboxylate or carbonyl groups (Figure 8D,E), along the analysed trajectory, besides its interactions with Glu/Asp on TM2. Interestingly, TM11′s arginine is relevant to substrate transport on 1B1/1B3 [26,36], which supports the relevance of Arg (TM11) with the ligand.

In OATP1B3 binding pose with E3S has shown interactions with Glu74, Asn77 and Arg580 (Figure 8E). In OATP2A1 binding pose with prostaglandin A2 shows approximately 30% of interactions over the simulation time. In OATP1A2, 1B3, 1C1, 2A1 ligands have shown water-mediated interactions with the Glu/Asp on TM2. These interactions are relevant to contextualizing earlier studies, where Glu74 has been reported to be important in substrate binding [25]. A more recent study also confirms the relevance of the interaction between Glu74/95 and Arg580/607 in OATP1B1 and 2B1 [32].

Last, the work of Tuerkova et al. (2021) [32] discusses OATP’s signature dynamics, such as the increased flexibility at the TM1/TM2 interface and to a lesser extent TM7/TM11 as relevant to determining the substrate binding. Their normal mode analyses were generated around a specific structural template (FucP transporter) to that the final conformational ensemble would cover and represent this movement. The conformation of TM1/TM2 was shown to affect the geometry of the N-terminal domain, which is consistent with our findings that TM1/TM2 can establish multiple salt bridges (TM7 or TM11) modifying the pocket conformation.

The use of a combination of machine learning and conformation ensemble for OATP docking yielded a significant improvement in the hit [37]. The original interpretation of these findings suggested that OATP selectivity is related to amino acid sequence changes, instead of conformation differences [32].

Our work, on the other hand, aimed to use unbiased simulations to analyse relevant conformational changes in selected conserved motifs, to generate a dynamic ensemble of novel OATP models. It is relevant to notice that longer simulations of OATP1A2 have shown conformational changes, predicting open and closed states of the transporter [36].

## 3. Materials and Methods

### 3.1. Homology Modelling and Validation

The sequences of all OATPs were retrieved from UniProtKB (Appendix A), and a homologue search was carried out using DELTA-BLAST against the non-redundant protein database (nrPDB). Two templates were selected based on the sequence identity and query coverage. Multiple sequence alignments of OATP’s with the templates were carried out in PROMALS3D server using default parameters. Further, the sequence–structure alignments were edited manually. The alignments were then imported to Modeller for the model generation using its AutoModel function. A total of 100 models were generated using Modeller and refined using GalaxyRefine2 web server [38]. Further, models were evaluated for their stereochemistry using the Ramachandran plot. A total of 100 models were generated for each OATP subtype using Modeller and inspected for the loop geometry. The models with the lowest DOPE (Discrete Optimized Protein Energy) score were selected for further refinement and evaluation. Refined models were checked for the quality using Ramachandran plot. (Appendix A). In these plots, all models showed more than 95% of residues in favoured regions indicating the model’s quality.

### 3.2. Model Evaluation

Ramachandran plots were used to check the quality of the models by plotting the dihedral angles of the backbone (phi and psi). Refinement was carried out using GalaxyRefine2 web server [38] which uses loop modelling, and overall relaxation methods to improve the quality of the models. The refined models were used to for docking with the known substrates.

### 3.3. Preparation of Ligands

Available data for substrates of all OATPs were collected from databases such as Metrabase [39] and CHEMBL [40]. All the structures were prepared by using LigPrep [41] module in Schrodinger package. Epik [42,43] was used to generate the ionization states at pH 7.0 (±2.0), and all stereoisomers were generated using OPLS3e [44] force field. Macromodel [45] conformational search was used to generate the conformers for all the prepared substrates with all default settings except for the method used Torsional Sampling (MCMM). Substrates were collected from Metrabase, prepared using LigPrep [41] and generated conformers using Macromodel. The Macromodel generated conformers were used as input for docking, allowing ligand flexibility and keeping the receptor rigid.

### 3.4. Preparation of Protein

The homology models were prepared using Protein preparation wizard [46] in Schrodinger package. Hydrogens were added; bond orders assigned and hydroxyl, Asn, Gln and His states optimized using ProtAssign at pH 7.0. The models were minimized while restraining the heavy atoms and converging to RMSD 0.3 Å and hydrogens unrestrained using OPLS3e forcefield.

### 3.5. Binding Site Analysis and Classification

We have extracted the residues around 8 Å from the centre of each OATP transporter surrounding the binding pocket in the pore, whose side chains are facing inwards. The residues were aligned according to the transmembrane location and position as seen in Figure 2A. The calculated identity and similarity matrices of these amino acids can be found in the Appendix A. Heatmap was generated using these identity and similarity matrices which in turn resulted into 3 classes of OATPs.

### 3.6. Sitemap and Grid Generation

SiteMap [47,48] was used to find top-ranked potential receptor binding sites, with at least 15 site points per reported site and up to 5 sites per protein. The residues around the top-ranked binding site were selected as grid centre with Receptor Grid generation [49] module in Schrodinger keeping all other settings as default. Sitemap has produced 4–5 sites for each OATP out of which the best ranked sites were selected for docking analysis with Dscore more than 0.8, which suggests good druggability and a site which is in the pore of the transporter. The grid was generated by selecting the centroid of the residues from the site region predicted by the sitemap.

### 3.7. Docking of Substrates

Due to low homology with the template proteins, homology models were further validated using docking. Glide [49,50] Standard precision (SP) with flexible ligand sampling was employed for docking of ligands. Epik state penalties were applied to docking scores for the ligands which adopted higher energy states. The van der Waals radii and partial charge cut-off for non-polar ligand atoms were set as 0.80 and 0.15, respectively, to soften the potential.

### 3.8. Molecular Dynamics

Desmond [51] was employed to build the system in orthorhombic boxes (buffer distance 13 Å in all directions) with protein immersed in TIP3P (transferable intermolecular potential with 3 points) solvation model and DMPC (dimyristoylphosphatidylcholine) membrane pre-equilibrated at temperature 300 K placed perpendicular to the central pore of protein keeping the helices inside the membrane. OPLS3e force field was used to build the system, and Na^+^/Cl^−^ ions were added to neutralize the system. Simulations were performed with isothermal-isobaric ensemble with constant particle number, pressure and temperature (NPT). The temperature was maintained at 300 K using a Nose–Hoover chain [52,53] thermostat, and the pressure was maintained at 1.01 bar with a Martyna–Tobias–Klein barostat. Default relaxation protocol was used to minimize and perform short simulations to relax the system. RESPA integrator with a time step of 2 fs for bonded terms, near (2 fs) and far (6 fs) was used. A cut-off radius for short-range coulombic interactions was set to 9 Å. The velocities of atoms were initialized using a random seed. Three replicates of each simulation for 200ns were run with random seeds of initial velocities for both apo and docked complexes.

### 3.9. Molecular Dynamics Analysis

The trajectories from the simulations were submitted to simulation interaction diagram (SID) calculations in a Schrodinger package. All the interaction frequency figures and bubble graphs were generated using the raw data generated from SID. Distance calculations were carried out using the trajectory_asl_monitor.py script from Schrodinger package. 

### 3.10. Data Visualization

Visualization of protein structures was carried out using PyMOL (The PyMOL Molecular Graphics System, Version 2.0 Schrödinger, LLC). 2D topology of OATP structure was created with BioRender.com (accessed on 8 July 2022). All the 2D chemical structures shown in Figure 1 were drawn and visualized using Chemdraw.

### 3.11. Electrostatic Potentials and Hydrophobic Volumes Calculation

Electrostatic potential surface maps were generated using the APBS electrostatics from PyMOL Molecular Graphics system version 2.5.1. PoreWalker server was used to identify the pore, diameter profiles at various heights throughout the pore and residues lining the channels [54]. Hydrophobicity was calculated using the Hydrophobic/philic surfaces tool in Schrodinger package (2022-2).

## 4. Conclusions

Organic anion transporting polypeptide (OATP) family transporters are involved in governing the transport of various endogenous substances and drugs. Despite their recognized role in the drug transport and drug delivery, OATPs structural details are not known yet. Towards this goal, the current study provides the OATP structural models that can be used to study the substrate-transport/drug-transport mechanisms. Our OATP models, originated from homology modelling and molecular dynamics simulations, has shown diverse conformations based on the binding of substrates. The models suggest that OATPs share a conserved transmembrane helix folding harbouring a druggable binding pocket within the inner pore. Positive electrostatic potential surfaces were observed in class 1, class 2 and OATP5A1 in class 3 OATP transporters. Hydrophobic surface volumes revealed the hydrophobic nature of class 2 and class 3 OATP transporters. Substrate docking studies displayed multiple potential binding modes, and simulations provided insights into the substrate movement from the external pore towards the cytoplasm resembling the substrate transport in some of the transporters such as OATP1A2. We have identified conserved salt bridges between glutamate on TM1 and lysine on TM7 which showed opening and closing conformations in the simulations of the OATP transporters and could play key role in the entrance of the substrates. In the OATP1 family, conserved lysine on TM1 interacts with conserved motif ‘RGI/MGE’ on TM4 and could be relevant in the binding and transport of ligands. In terms of structural elements, the TM1/TM2 acts as a core element by interacting with two novel conserved motifs in the TM7 and TM11. In our simulations, TM2-TM7 interaction seems to influence the inner pocket accessibility, while TM2-TM11 salt-bridges control the substrate binding stability. This study also focused on using models to interpret the single point mutation data for a small set of transporters, and therefore, a more comprehensive study is needed, discussing the substrate-binding mechanisms.

## Figures and Tables

**Figure 1 molecules-27-08531-f001:**
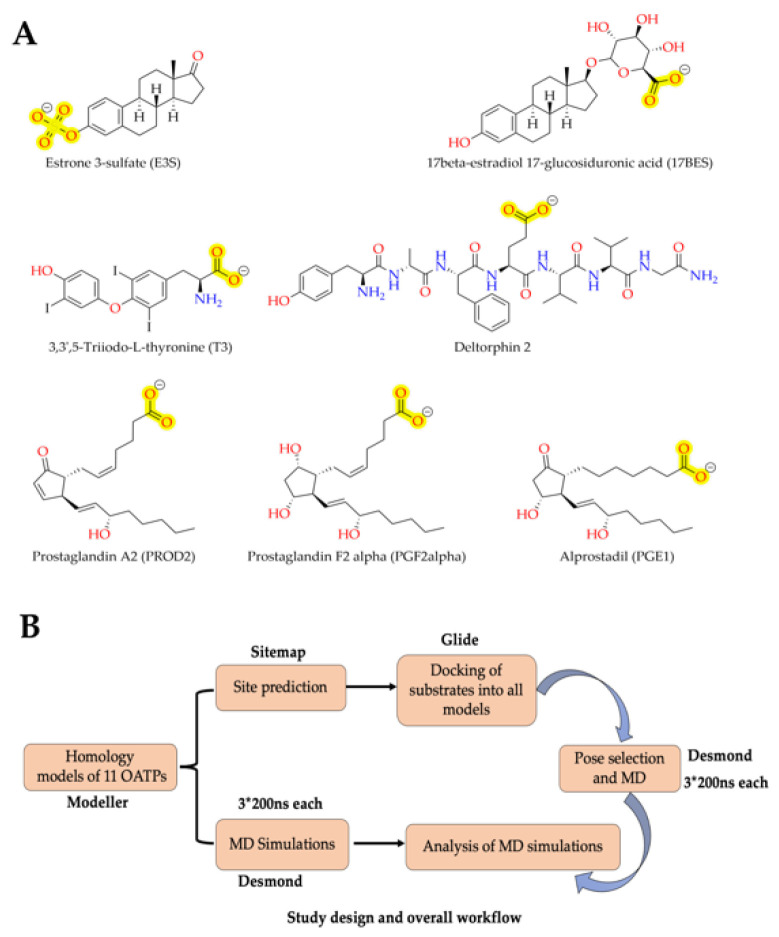
(**A**) 2D structure representation of the compounds OATP ligands employed in this study, negatively charged anions are highlighted in yellow. (**B**) Overall study design and workflow followed.

**Figure 2 molecules-27-08531-f002:**
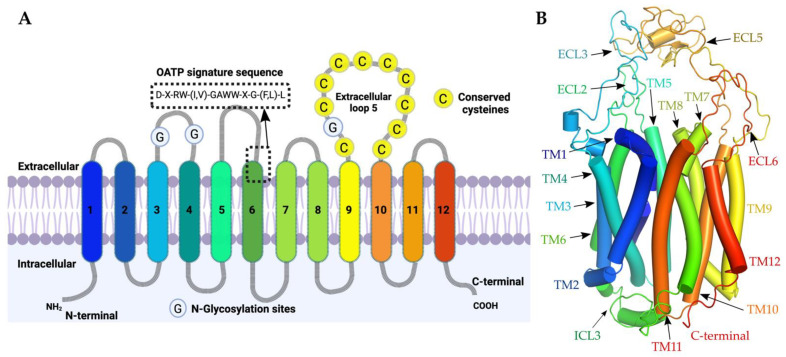
(**A**) OATP 2D model with transmembrane helices placed in membrane with labels. Created with BioRender.com (accessed on 8 July 2022). (**B**) A representative OATP 3D homology model with minimized in-ward open conformation.

**Figure 3 molecules-27-08531-f003:**
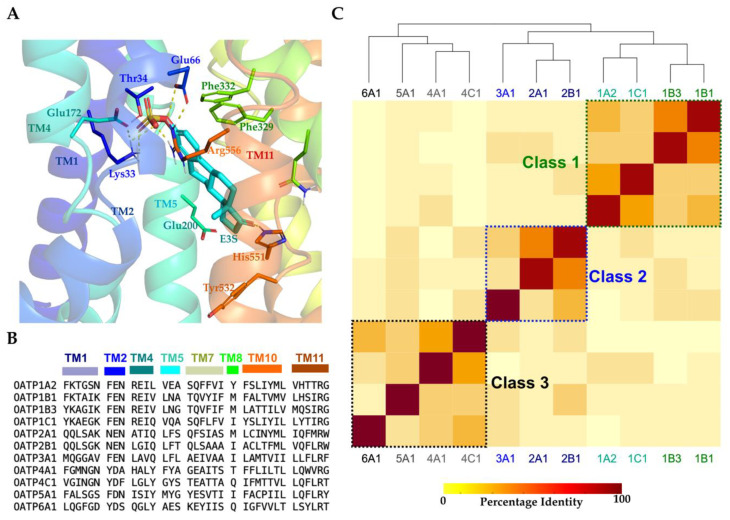
(**A**) Representative potential binding site (OATP1A2 shown here) predicted using SiteMap. transmembrane helices are colored in rainbow colors and the ligand is shown in cyan sticks representation. (**B**) Binding site residues alignment along the pore shown in OATPs. Helix colouring is displayed above the sequences (**C**) Heatmap showing clusters generated from identity matrix.In the matrix warmer colors for example dark red indicate high percentage identity and colder colors indicate lower percentage identity.

**Figure 4 molecules-27-08531-f004:**
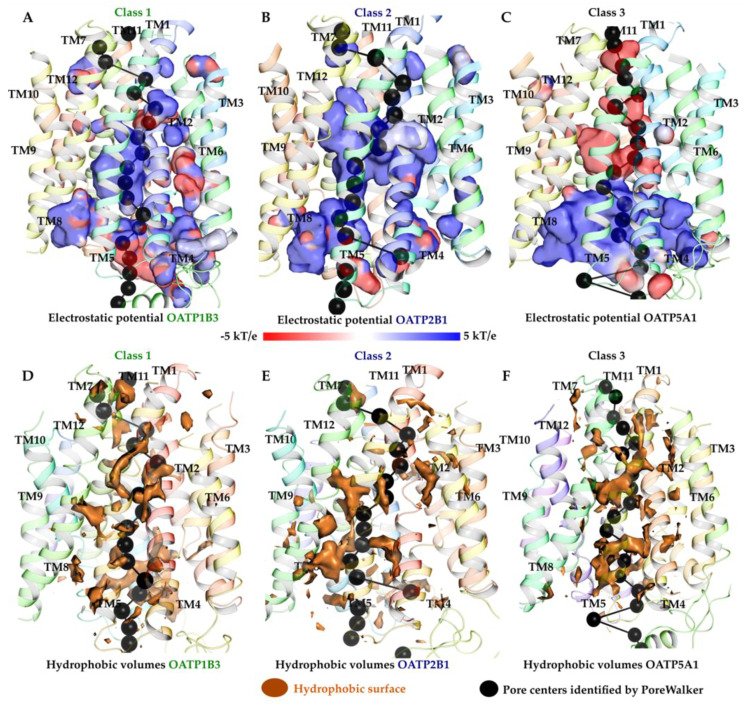
Electrostatic potential surfaces (**A**–**C**) and hydrophobic volumes (**D**–**F**) shown along the pore. (**A**) Electrostatic potential surfaces of class 1 (OATP1B3). (**B**) Electrostatic potential surfaces of class 2 (OATP2B1). (**C**) Electrostatic potential surfaces of class 3 (OATP5A1). Positive potentials shown in blue, negative in red, neutral in white and hydrophobic surfaces in orange.

**Figure 5 molecules-27-08531-f005:**
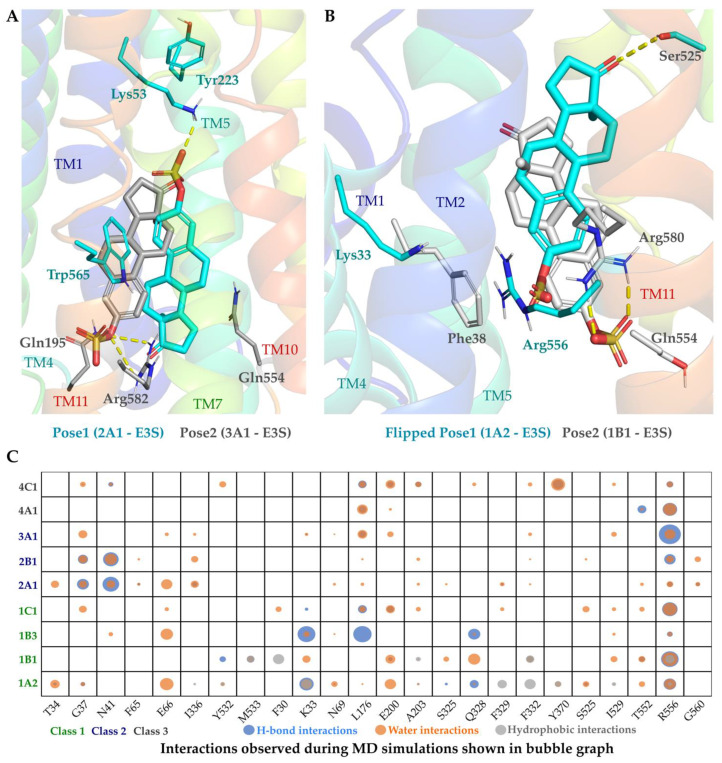
(**A**) Representative binding poses depicting Pose1 (OATP2A1) and Pose 2 (OATP3A1) observed in the docking. (**B**) Binding pose flip observed in OATP1A2 (Cyan) compared to OATP1B1 (grey) where there is a change from Pose1 to Pose2. (**C**) Frequency of protein–ligand interactions along the molecular dynamics simulation trajectories are shown in bubble graph; the residues are named according to OATP1A2, and the corresponding aligned residues could be referred from Figure 3B. All the docking poses shown above are of E3S.

**Figure 6 molecules-27-08531-f006:**
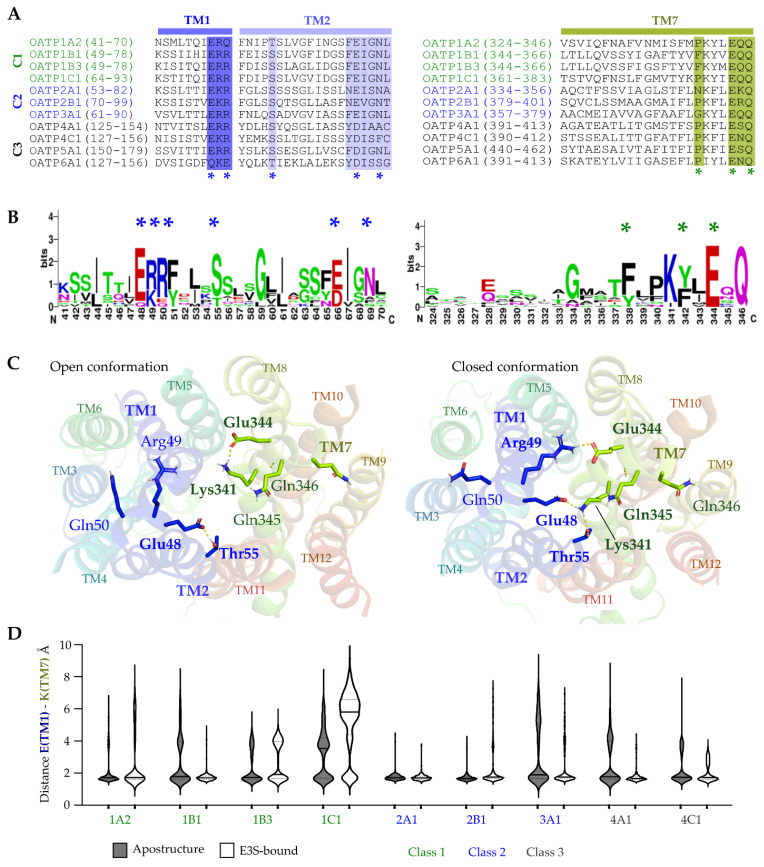
(**A**) Alignment of OATPs showing TM1 and TM2 highlighting the important residues in blue (left) and TM7 (right) with residues involved in the interactions with TM1 displayed in green. (**B**) HMM-Logos showing the conservation of highlighted residues. (**C**) Salt bridges involved in the opening (left) and closing (right) of the entrance of the transporter (displayed on the OATP1A2). (**D**) Distance between Glutamate on TM1 and Lysine on TM7 observed in the molecular dynamics simulations. Conserved amino acids are indicated with * in A and B.

**Figure 7 molecules-27-08531-f007:**
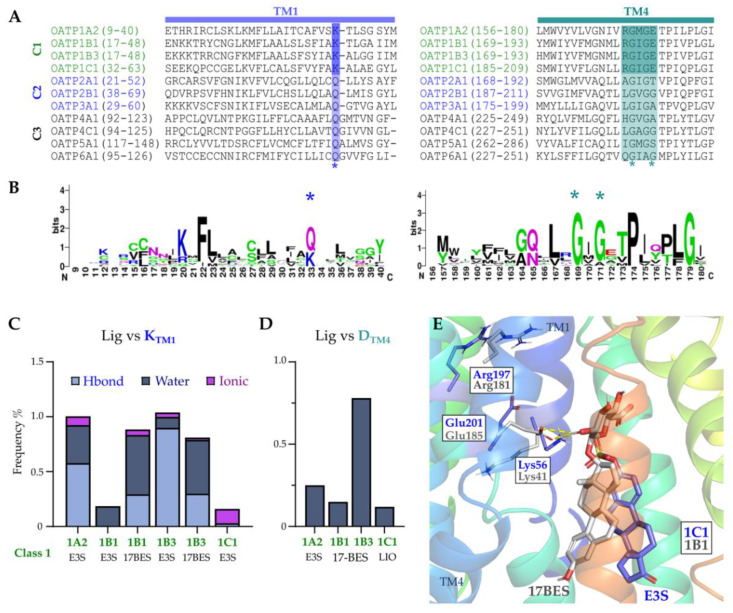
(**A**) Alignment of OATPs showing TM1 (left) highlighting the Lysine residue in violet and TM4 (right) highlighting the Arginine and glutamate. (**B**) HMM-logos showing the conservation of highlighted residues. (**C**) Interaction frequency of ligands with conserved Lysine on TM1. (**D**) Interaction frequency of ligands with Glutamate on TM4 (water interactions). (**E**) Binding pose of 17BES in OATP1B1 (grey) and E3S in OATP1C1 (violet) showing interactions with Glu185 and Lys56, respectively. Conserved amino acids are indiacted with * in A and B.

**Figure 8 molecules-27-08531-f008:**
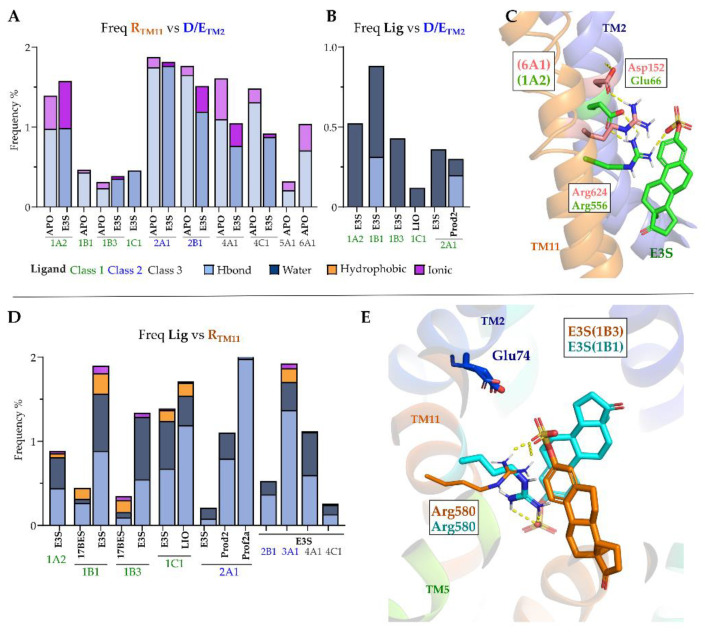
(**A**) Interactions between Arginine on TM11 and Glutamate or Aspartate on TM2 observed during MD simulations. (**B**) Interaction frequency of ligands with Glu/Asp on TM2. (**C**) Representation of interactions between Arginine (TM11) and Glu/Asp (TM2) in OATP1A2 (green) and OATP6A1 (light pink). (**D**) Interaction frequency of ligands with Arginine on TM11. (**E**) Binding poses of E3S in OATP1B1 (cyan) and OATP1B3 (orange) showing interactions with Arginine on TM11.

**Table 1 molecules-27-08531-t001:** Results from the delta blast with percentage identities between the query and template protein sequences.

OATP	Uniprot ID	Amino Acid Length	Gene Name	DELTA-BLAST Search		Alignment Used for Model Building	
				% Identity with 6E9N	% Identity with 1TGS	% Identity with 6E9N	% Identity with 1TGS
OATP1A2	P46721	670	SLCO1A2	11.4	30.2	13	27.3
OATP1B1	Q9Y6L6	691	SLCO1B1	11.9	21	15.2	27.3
OATP1B3	Q9NPD5	702	SLCO1B3	11.6	21	14.4	27.3
OATP1C1	Q9NYB5	712	SLCO1C1	13.5	27.9	14.2	25.5
OATP2A1	Q92959	643	SLCO2A1	12.3	30.4	14.2	40
OATP2B1	O94956	709	SLCO2B1	11.8	18.7	12.2	21.8
OATP3A1	Q9UIG8	710	SLCO3A1	12	27.2	15	29.1
OATP4A1	Q96BD0	722	SLCO4A1	9.7	24.1	11.5	25.5
OATP4C1	Q6ZQN7	724	SLCO4C1	10.7	28	13.7	27.3
OATP5A1	Q9H2Y9	848	SLCO5A1	11.5	31.5	13	32.7
OATP6A1	Q86UG4	719	SLCO6A1	10.9	26	13.7	29.1

## Data Availability

All the files of alignments, homology models, docking poses and molecular dynamics simulation trajectories have been deposited on Zenodo web repository, and the link can be found here: https://zenodo.org/record/7071180 accessed on 12 September 2022.

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
