# Peer review of "Comparative Modelling of Organic Anion Transporting Polypeptides: Structural Insights and Comparison of Binding Modes"

_molecules, 2022, doi:10.3390/molecules27238531_

Round 1

Reviewer 1 Report

The manuscript entitled “Comparative Modelling of Organic Anion Transporting Polypeptides (OATPs): Structural Insights and Comparison of Binding Modes” used a combination of homology model with molecular dynamics simulations to generate a comprehensive structural dataset, that encompasses a diverse set of OATPs but also their relevant conformations. The OATP models share a conserved transmembrane helix folding harboring a druggable binding pocket in the shape of an inner pore. The simulations suggest that the conserved salt bridges at the extracellular region between residues on TM1 and TM7 might influence the entrance of substrates.  However, the presented article could be published in Molecules after major revision for the following:

1.     It is better to remove abbreviations from the article title.

2.     Table T1: Results from the delta blast with percentage identities between the query and template protein sequences. It is better to transfer to the original manuscript

3.     Correct reference number 9. It is recommended to revise all references manually as there are many errors happened by reference programs such as endnote, Mendeley, etc.

4.     The figure of merits and limitations of the proposed binding model should be presented.

Author Response

Reviewer 1: The manuscript entitled “Comparative Modelling of Organic Anion Transporting Polypeptides (OATPs): Structural Insights and Comparison of Binding Modes” used a combination of homology model with molecular dynamics simulations to generate a comprehensive structural dataset, that encompasses a diverse set of OATPs but also their relevant conformations. The OATP models share a conserved transmembrane helix folding harboring a druggable binding pocket in the shape of an inner pore. The simulations suggest that the conserved salt bridges at the extracellular region between residues on TM1 and TM7 might influence the entrance of substrates.  However, the presented article could be published in Molecules after major revision for the following:

  1. It is better to remove abbreviations from the article title.

Response: We have now removed the abbreviation from the title.

  1. Table T1: Results from the delta blast with percentage identities between the query and template protein sequences. It is better to transfer to the original manuscript

Response: Thanks for your comment, this data has been transferred to the main text in the manuscript (now it is placed on page 4, Table T1).

  1. Correct reference number 9. It is recommended to revise all references manually as there are many errors happened by reference programs such as endnote, Mendeley, etc.

Response: Thanks for the suggestion, we checked the references accordingly and adapted.

  1. The figure of merits and limitations of the proposed binding model should be presented.

Response: We strongly agree with the reviewer that discussing the limitations of the proposed binding mode is necessary. However, we have already added a paragraph pointing these out by the end of the discussion, instead of using a figure for this.

Reviewer 2 Report

The structural analysis of OATP (organic anion transporting polypeptide) based on homology modeling and molecular dynamics simulation is admirable. The authors built the OATP atomic model using the homology modeling method, then they stimulated drug binding based on those models. Their findings will be useful when designing drugs that target OATP, but I have some reservations.

Due to the high accuracy of alphafold2, the authors may modify their homological model to the predicted model, or compare their model to the predicted model.

The OATP pore plays a critical role; therefore, it is better to illustrate and analyze it first. For example, using the molecule surface to display the pore; measuring the size of the pore; calculating the hydrophobicity and electrostatic potential for the pore.

It is possible that the 5th extracellular loop functions as the gate of the OATP pore. This loop should be modelled with a template with the same function. Furthermore, since there are 10 cysteines in this loop, I wondered if disulfide formation is occurring, and if so, it might be regulating the pore.

There is no obvious comparison between the open and closed conformations in Figure 5C. It is better to provide an overview of the pores in two conformations, then proceed to the details.

In line 154, “based on the multiple sequence alignment and literature.” It is better to cite the literature here.
